# Broadband Supercontinuum Generation in Dispersion Decreasing Fibers in the Spectral Range 900–2400 nm

**Irina V. Zhluktova** [1,2], **Vladimir A. Kamynin** [1,*], **Dmitry A. Korobko** [2], **Aleksei S. Abramov** [2], **Andrei A. Fotiadi** [3,4], **Alexej A. Sysoliatin** [1] **and Vladimir B. Tsvetkov** [1]

1   Prokhorov General Physics Institute of the Russian Academy of Sciences, 38 Vavilov Street, 119991 Moscow, Russia
2   S.P. Kapitsa Scientific Technological Research Institute, Ulyanovsk State University, 42 Leo Tolstoy Street, 432970 Ulyanovsk, Russia
3   Electromagnetism and Telecommunication Department, University of Mons, 7000 Mons, Belgium
4   Finland Opto-Electronics and Measurement Techniques, Faculty of Information Technology and Electrical Engineering, University of Oulu, Erkki Koiso-Kanttilankatu 3, 90570 Oulu, Finland
*   Correspondence: kamynin@kapella.gpi.ru

**Abstract:** The spectrally flat supercontinuum generation in the wavelength range of 900–2400 nm is demonstrated in silica-based fibers of variable core diameter and dispersion. It is shown that, in comparison with standard optical fibers of the same length, supercontinuum spectra 200 nm wider can be realized in the samples under study. The significant difference between the spectral and temporal transformations of radiation depending on the direction of propagation is demonstrated in the researched fiber samples.

**Keywords:** all-fiber supercontinuum generator; ytterbium-doped fiber laser; dispersion decreasing fiber; octave-spanning supercontinuum; fiber amplifiers

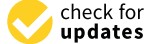



## 1. Introduction

In recent decades, supercontinuum (SC) generation has become the subject of research for many scientific groups since this topic is of interest in various fields, both scientific [1] and practical. For example, for remote sensing [2], respiration analysis [3], spectroscopy and residual gas detection [4], hyperspectral microscopy [5], early diagnosis of diseases [6], coherence tomography [7], and optical communication [8]. Separately, in some cases used in time-frequency metrology based on ultra-stable femtosecond optical frequency combs [9].

One of the most common schemes for obtaining ultra-wide optical spectra is the use of a master oscillator (MO) in a pulsed mode with the subsequent amplification of radiation which is introduced into a nonlinear medium, where the optical spectrum is transformed. The broadening occurs due to the joint action of a number of nonlinear effects (four-wave mixing (FWM), self-phase modulation (SPM), stimulated Raman scattering (SRS), soliton self-frequency shift, etc.) on the propagating radiation [10–12]. To achieve the greatest wavelengths of the red boundary of SC generation, laser systems emitting in the spectral range of 1.5–2 μm are used as MO. However, it is known that in the optical spectra of SC generators based on standard silica fibers, the long-wave edge is cut short in the region of 2.5 μm due to the fact that silicon dioxide has a relatively high level of phonon energy ~1100 cm$^{-1}$ [13], which blocks the way for generating SC in the mid-infrared range. Therefore, for these media, the use of laser sources with generation wavelengths of 1–1.1 μm is prospective to achieve the maximum width of the optical spectrum due to the short-wavelength range and obtaining an entire octave [14].

Nowadays, fibers with a high nonlinearity coefficient are increasingly used to generate SCs, such as tellurite, fluoride, or chalcogenide [15–18], etc. Also, silica fibers doped with germanium oxide (GeO$_2$) [19–23] are used as a nonlinear medium, one of the types of

which are dispersion decreasing fibers (DDF) [24,25]. This type of fiber is made on a silica basis, which facilitates their integration into a laser system based on standard fibers. This circumstance helps to avoid the use of additional optical elements [26,27] for the input/output of radiation into and out of a nonlinear medium. Moreover, the use of DDF fibers provides efficient radiation conversion due to the compression of the soliton pulse, which will be located in the zone with anomalous dispersion at first and then move to zero [28]. And this triggers the SRS self-frequency shift to allow the pulse to be back to the anomalous dispersion [29]. The combination of these effects helps to realize an effective conversion of radiation in a wide spectral range when using DDF. In fibers with an optimal dispersion profile along the length, a coherent continuum is generated with an unevenness of several dB, symmetrical to pumping and, most importantly, resistant to spontaneous emission noise. It is significant for promising communication systems Optical CDMA & OAWG (Optical Arbitrary Waveform Generation) [30,31].

The article presents an all-fiber supercontinuum generator based on a pulsed MO in combination with a fiber amplifier and nonlinear fibers (NF). The output parameters between the NF samples were also compared to obtain the most efficient SC generation.

## 2. Experimental Setup

Figure 1 shows the optical scheme of the SC generator. The source was built according to the standard scheme: MO + fiber amplifier + nonlinear medium. The use of silica-based fibers made it possible to build a complete fiber circuit without mechanical splices and volumetric optics.

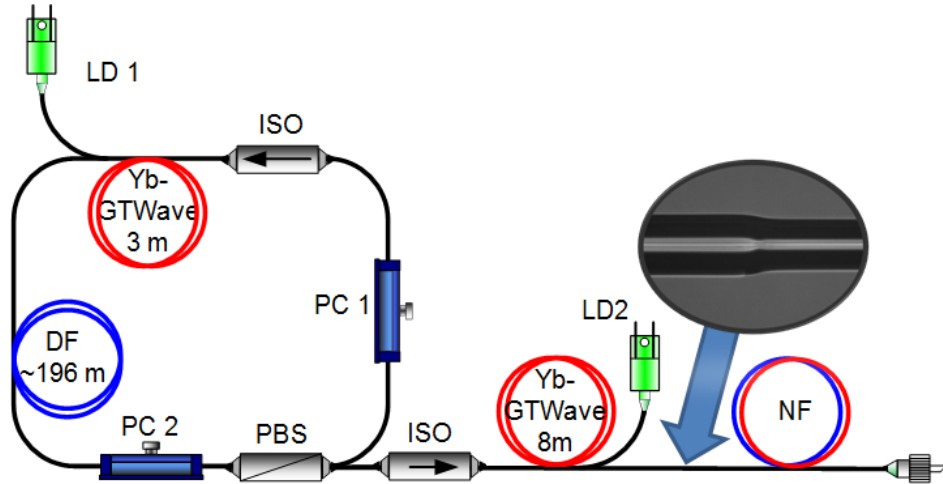

**Figure 1.** Experimental setup: ISO—optical isolator, LD1,2—pump laser diodes, PC1,2—polarization controllers, DF—delay fiber, PBS—polarization beam splitter, NF—nonlinear fiber.

### 2.1. MO and Amplifier

An all-fiber ytterbium-doped (Yb) laser operating in the passive mode-locking regime based on nonlinear polarization evolution (NPE) was used as the MO. A ytterbium-doped multicomponent fiber (GTWave) [32] was used as the active laser medium, the active part of which was doped with ytterbium ions. The core diameter of this fiber was 6 μm with cladding to 125 μm and an active impurity concentration of $8.5 \times 10^{-19}$ cm$^{-3}$. The absorption at the pump wavelength (λ = 976 nm) was approximately 1.06 dB/m. The numerical aperture for the active fiber with a core and cladding was 0.11 and 0.45, correspondingly. The active fiber of the Yb laser was pumped by a multimode semiconductor laser diode emitting at a wavelength of 976 nm with an output power of up to 1.5 W, the fiber output of which was subsequently connected to the passive part of the GTWave fiber. The total length of the Yb laser cavity was about 200 m, which corresponded to a fundamental pulse repetition rate of 1 MHz and pulse duration of 260 ps (Figure 2a) [33]. The maximum intensity value of the laser output radiation spectrum (Figure 2a) corresponded to a wave-

length of 1063 nm, and the spectrum shape corresponded to a strongly chirped dissipative soliton [34]. At the output of the MO, an average power of 5 mW was obtained, and the pulse energy was 5 nJ. Also, there was a peak at 1120 nm in the spectral range related to the 1st Stokes component.

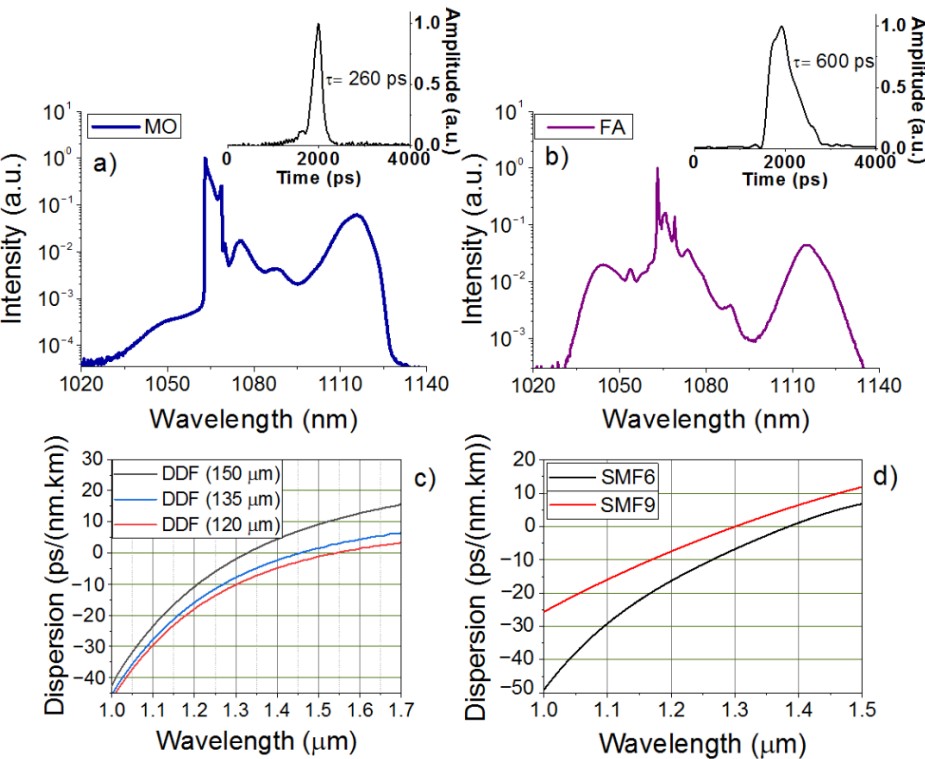

**Figure 2.** The output parameters of the MO (**a**) with subsequent conversion of its characteristics after using a fiber amplifier (**b**), as well as graphs of the dispersion for different diameters of DDF (**c**) and SMF (**d**).

A ytterbium-doped fiber amplifier (YDFA) was used to increase the power required for efficient SC generation. The active medium was a Yb GTWave fiber 8 m long (absorption @976 nm—9.6 dB/m, active impurity concentration $10.2 \times 10^{-19}$ cm$^{-3}$), which was pumped at the wavelength of 976 nm by a semiconductor laser diode (power up to 8 watts). A fiber isolator was placed between the MO and YDFA to prevent reflected and scattered radiation from entering the Yb laser. Primary broadening and transformation of the MO optical spectrum took place in YDFA. Precisely, a significant increase in the peak widening at wavelengths of 1040 and 1120 nm due to SRS with a transformation in the region of 1063 nm (Figure 2b) was caused by the SPM effect. The average output power was changed from 5 mW to 800 mW, and the pulse duration increased from 260 ps to 600 ps with an increase in energy to 800 nJ (0.8 μJ), as it is shown in Figure 2b.

## 2.2. Researched Fibers

Various fiber samples were used as nonlinear media for the conversion of amplified radiation, such as DDF and standard single-mode fibers (SMF). Table 1 shows the main parameters of the studied samples. The DDF fiber sample exhibited an anomalous dispersion that varied linearly along the length of the fiber. This sample of fiber was produced in the course of joint research of GPI RAS and IHPS RAS using the classical technique of taper-type fiber drawing while maintaining a smooth change in the core diameter (from 6 to 9 μm) along the length of the entire sample in accordance with some prearranged low [35,36]. Therefore, SMFs with different core diameters, 6 and 9 μm, were selected for comparison. The SMF lengths were matched to the DDF sample. Separately, Figure 2c,d shows the dispersion curves for the DDF and SMF samples.

**Table 1.** Parameters of used NF samples.

| Sample | d1 [1], μm | d2 [1], μm | L [2], m | α [3], dB/km |
|--------|--------|--------|-------|----------|
| DDF | 6 (120) | 9 (150) | 78 | 0.9 |
| SMF6 | 6 | 6 | 78 | 0.2 |
| SMF9 | 9 | 9 | 78 | 0.2 |

[1] core diameter. [2] fiber length. [3] optical losses.

The DDF fiber used in this work was silica-based, doped with germanium oxide, which made it possible to provide a spliced connection with the elements of the optical system (as shown in Figure 1—A photograph of the splicing of the laser system with a DDF sample). This fiber was placed after the fiber amplifier and was connected using a fusion splicer Fujikura 86S. The optical spectra of the output radiation from the MO, YDFA and NF were obtained using spectrum analyzers operating in the spectral ranges of 600–1700 nm and 1200–2400 nm with a resolution of 0.5 nm. An optical complex with an ultrafast photodiode with a rise time of 35 ps and oscilloscopes with a bandwidth up to 4 GHz was used to estimate the pulse duration.

## 3. Results and Discussion

### 3.1. Standard Fibers versus DDF

For the efficient generation of SC, it was necessary to select a suitable NF due to the use of which super-broad radiation would be obtained. In this research, we used three samples: DDF and two standard SMFs with different core diameters. SMFs were selected as references because of the previously demonstrated results of SC generation [37–40] in a wide spectral range.

When using SMF samples, ultra-broad radiation was obtained: for SMF6, the maximum width of the optical spectrum was 895 nm (Figure 3a) since the radiation conversion goes from 1 μm to 1.8 μm, and for SMF9—1212 nm (Figure 3c), with a conversion in the region of 1–2.1 μm. In both cases, the width was estimated by a signal level of −30 dB (as shown in the Figure 3, $10^{-3}$). The average output power for these samples is 300–400 mW, and the estimated envelope pulse duration is 500–700 ps, which is commensurate with the duration of the radiation converted after YDFA.

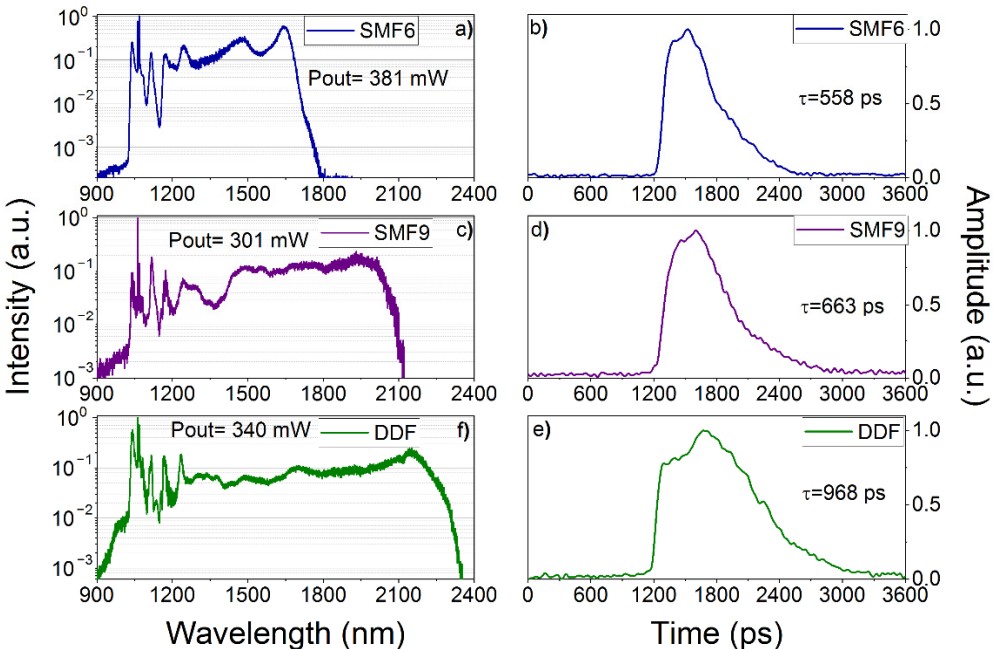

**Figure 3.** SC optical spectra (**a**,**c**,**f**) were obtained when using various NF samples and pulse envelopes at an input energy of 0.8 μJ (**b**,**d**,**e**).

In comparison with the SMF samples, when using the DDF sample, it can be seen that a transformation appears not only in the long-wavelength region of the spectrum (in the region of 2 μm) but also in the short-wavelength region (down to 900 nm). Furthermore, the spectrum demonstrates a transition from some dominant nonlinear effects (SRS in the wavelength range 1111 nm (8998 cm$^{-1}$), 1163 nm (8598 cm$^{-1}$), 1220 nm (8198 cm$^{-1}$)) to others (the soliton generation in the spectral region 1500–2100 nm) due to the transition from normal to anomalous dispersion [41]. Table 2 presents the obtained parameters of the SC depending on the samples used. We can see that the most effective generation was obtained while using DDF with the maximum optical spectrum width of 1496 nm at the signal level of −30 dB. The envelope pulse duration is longer when using a DDF, and it reaches up to 1 ns (Table 2 and Figure 3e).

**Table 2.** The comparison of output parameters depending on the sample used.

| Sample | Δλ [1] (−30 dB), nm | Long-Wave Length Limit, nm | $P_{av}$ [2], mW | τ [3], ps |
|---|---|---|---|---|
| DDF (120–150 μm) | 1496 | Up to 2354 | 340 | 970 |
| DDF (150–120 μm) | 1355 | Up to 2246 | 324 | 730 |
| SMF6 | 895 | Up to 1810 | 381 | 560 |
| SMF9 | 1212 | Up to 2100 | 301 | 660 |

[1] spectral width. [2] average output power. [3] pulse envelope duration.

### 3.2. SC Generation Depending on Propagation Direction in DDF

After selecting the most efficient SC sample for generation, experiments were carried out on the effect of the DDF fiber position on the ultra-broad emission. Namely, this sample was introduced into the laser system not only from a smaller diameter to a larger one (120–150 μm) but also vice versa; it was deployed (150–120 μm) in order to study the transformation depending on the propagation along the sample (Figure 4). Several control points were selected for the pumping power of the amplifier to compare the optical spectra (E = 0.149 μJ, E = 0.48 μJ, E = 0.8 μJ). It can be seen that when using a fiber sample with increasing diameter, the spectrum is transformed in the long-wavelength region, and the edge of the spectrum is shifted beyond 2.3 μm. If the sample is used in the back direction, then the transformation takes place already in the short-wavelength region, namely, after 900 nm and goes into the visible spectral range. The fiber orientation did not affect the average output power, which varied at the maximum pumping power of the amplifier.

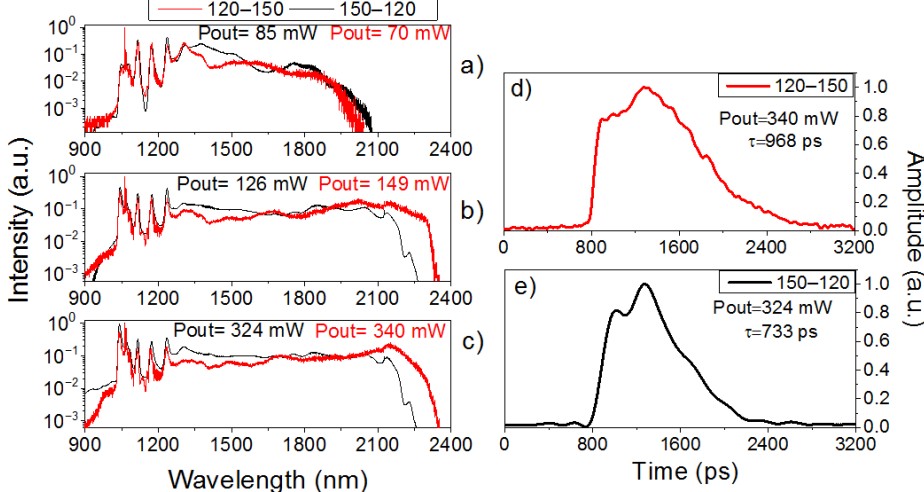

**Figure 4.** The comparison of the SC spectra obtained at the same amplifier pump powers for the DDF sample in dependence on the propagation direction ((**a**) E = 0.149 μJ, (**b**) E = 0.48 μJ, (**c**) E = 0.8 μJ), and envelopes of the obtained pulses (**d**,**e**).

The durations of the envelope pulses for both options were also obtained and are presented in Figure 4d,e. The maximum duration was in the region of 970 ps with maximum pumping of the amplifier (120–150 μm), and for the sample version 150–120 μm, the duration did not exceed 730 ps.

A feature of cone-type fibers is that the propagation of radiation is accompanied by a continuous increase in the propagation angle. Since this fiber is close to the cone-shaped fibers, some rays leave the cone through the side surface due to a violation of the conditions of total internal reflection. The active core absorbs part of the input power, the other part exits through the output end, and the remaining part leaves the cone through the side surface almost along its entire length due to violation of the conditions of total internal reflection. Thus, by changing the orientation of the DDF, we can effectively generate the SC both in the short-wavelength spectral range (Figure 4, black curve) and, vice versa, in the long-wavelength spectral range (Figure 4, red curve) since waveguide dispersion takes effect here, which affects the propagation of the mode in this fiber.

If we compare our results with other all-fiber silica-based supercontinuum generators, we are the first to demonstrate the possibility of generation in different spectral regions due to the orientation of the DDF fiber. Also, unlike many other systems, we have quite a small peak power input to NF for SC generation (1–2 kW vs. ~10 kW) [42,43], and the system is fairly simple and without complex elements (for example, in this paper, to get a full fiber circuit, the authors used a mode field adapter (MFA) [27]). If we compare with works using $GeO_2$ fibers [19–22], however, in our case, we go to the short-wave region of the spectrum due to the use of MO emitting in the 1 μm region and the taper of the DDF. To increase the energy in the pulse, further optimization of the studied system is required, as well as a more detailed analysis of the temporal dynamics of the pulses.

### 3.3. Numerical Simulation of SC Generation

For a better understanding of the processes which occur during the generation of an SC, the numerical simulation of the radiation propagation in the fibers used in the experiment was carried out. The well-known approach was used for calculations, using the generalized nonlinear Schrödinger equation for the field amplitude $A(z, t)$, which takes into account higher orders of dispersion (up to the eighth order $k \leq 8$) and Raman scattering [10,44,45]:

$$\frac{\partial A}{\partial z} - \sum_{k \geq 2} \frac{i^{k+1}}{k!} \beta_k(z) \frac{\partial^k A}{\partial t^k} = i\gamma \left(1 + i\frac{1}{\omega_0}\frac{\partial}{\partial t}\right)\left(A \int_{-\infty}^{\infty} R(t')|A(z, t - t')|^2 dt'\right). \quad (1)$$

Here $\omega_0$ is the carrier frequency corresponding to the wavelength $\lambda = 1.06$ μm,

$$R(t) = 0.82 \cdot \delta(t) + 0.18 \cdot \frac{\tau_1^2 + \tau_2^2}{\tau_1 \tau_2^2} exp\left(-\frac{t}{\tau_2}\right) sin\left(\frac{t}{\tau_1}\right)\Theta(t). \quad (2)$$

is the Raman response function, where the parameters $\tau_1$—12.2 fs, $\tau_2$—32 fs correspond to the response of the silica fiber, $\Theta(t)$ and $\delta(t)$ are the Heaviside function and the delta function, respectively. The known parameters of the fibers were used in the simulation (Table 1). The dispersion parameters $(\beta_k(z))$ of SMF6 and SMF9 fibers were obtained by approximating the dispersion curves (Figure 2d). For a fiber with a variable diameter, the dispersion parameters $\beta_k(z)$ were determined for the diameters indicated in Figure 2c and taken as reference (for example, for $d = 120$, $\beta_2 = 10.603$ $ps^2km^{-1}$, $\beta_3 = 0.008669$ $ps^3km^{-1}$, $\beta_4 = -1.632 \times 10^{-6}$ $ps^4km^{-1}$, $\beta_5 = 5.4432 \times 10^{-9}$ $ps^5km^{-1}$, $\beta_6 = 5.448 \times 10^{-11}$ $ps^6km^{-1}$, $\beta_7 = 2.1192 \times 10^{-13}$ $ps^7km^{-1}$, $\beta_8 = 2.7972 \times 10^{-16}$ $ps^8km^{-1}$). For intermediate values $d(z)$, the dispersion parameters were calculated by quadratic interpolation. Due to the fact that it is necessary to take into account a large amount of data in both time (~1 ns) and spectral (more than $10^{15}$ $s^{-1}$ wide) domains, the problem presents significant computational difficulty. To reduce computational costs, we limited the consideration to the initial conditions in the form of a frequency-modulated Gaussian pulse $A = \sqrt{P}exp\left(-(1 + iC)\frac{t^2}{2\tau^2}\right)$

with the duration of 100 ps, the peak power of $P = 1225$ W, and the chirp $C = 10^3$. The simulation window was $2^{18}$ grid points with $\Delta t = 3$ fs.

Initially, we will consider the propagation of radiation in fibers of constant diameter—SMF6 and SMF9. The evolution of the spectrum in fibers is shown in Figure 5. The characteristic feature of spectral evolution is a consistent multiple SRS, which leads to the transfer of energy towards longer wavelengths.

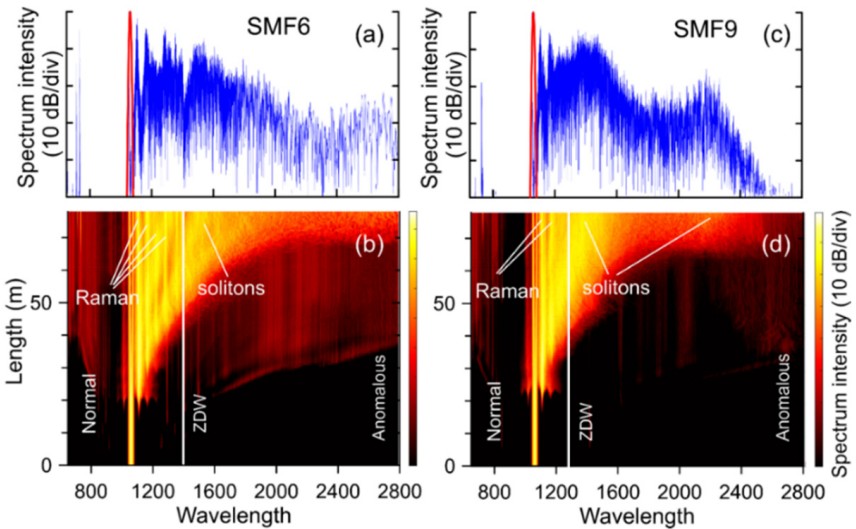

**Figure 5.** Input (red) and output (blue lines) spectra from (**a**) SMF6 and (**c**) SMF9 fibers. The evolution of the spectral density during the propagation of radiation along the fiber SMF6 (**b**) and SMF9 (**d**).

Ultrashort soliton pulses with a peak power of several kilowatts are formed from a part of the radiation that has crossed the wavelength of zero dispersion (ZDW) and passed into the region of anomalous dispersion. Under the action of SRS, they are shifted further toward the long-wavelength part of the spectrum. The situation in the time domain is illustrated in Figure 6. In changes of the pulse envelope, there are two mainly involved nonlinear processes. These are Raman scattering in the region of normal dispersion and the soliton formation in the anomalous dispersion region. One can see the optical wave-breaking process beginning near the pulse top, where the nonlinear effects are strongest. Initially, there is a Raman shift to longer wavelengths. The Figure 6 shows two consecutive Raman cascades. In the output pulse, traces of this process are noticeable near the trailing (right) edge of the pulse as sections with flat frequency modulation. Further, when a part of the pulse envelope passes into the region of anomalous dispersion (compared with Figure 5), the ultrashort solitons are formed, shifting to the range of longer wavelengths. In the output pulse, the group of solitons with the maximum peak power is shifted to the leading (left) edge. Frequency modulation (Figure 6a,d) shows that this group is spectrally shifted to the region of maximum wavelengths, which is consistent with the results of spectral evolution simulations (Figure 5). Comparing the SMF6 and SMF9 fibers, it can be seen that in the latter case (SMF9), due to the lower ZDW, the formation of solitons occurs earlier, and their final Raman shift is larger than for the SMF6 fiber. As can be seen, the smoothed envelope of the output pulse with a steep leading and flat trailing edge corresponds to experimental observations. Some differences between the simulation results and the experiment can be explained by the difference in the form of the initial signal, inaccuracies in the specification of fiber parameters, approximation of frequency independent nonlinearity coefficient and neglecting the frequency dependence of linear losses.

Finally, let us consider the propagation of radiation in DDF. The spectrum evolution is shown in Figure 7. As one can see, during evolution, the same discussed earlier processes occur, including multiple Raman transformations and the generation of ultra-short Raman solitons.

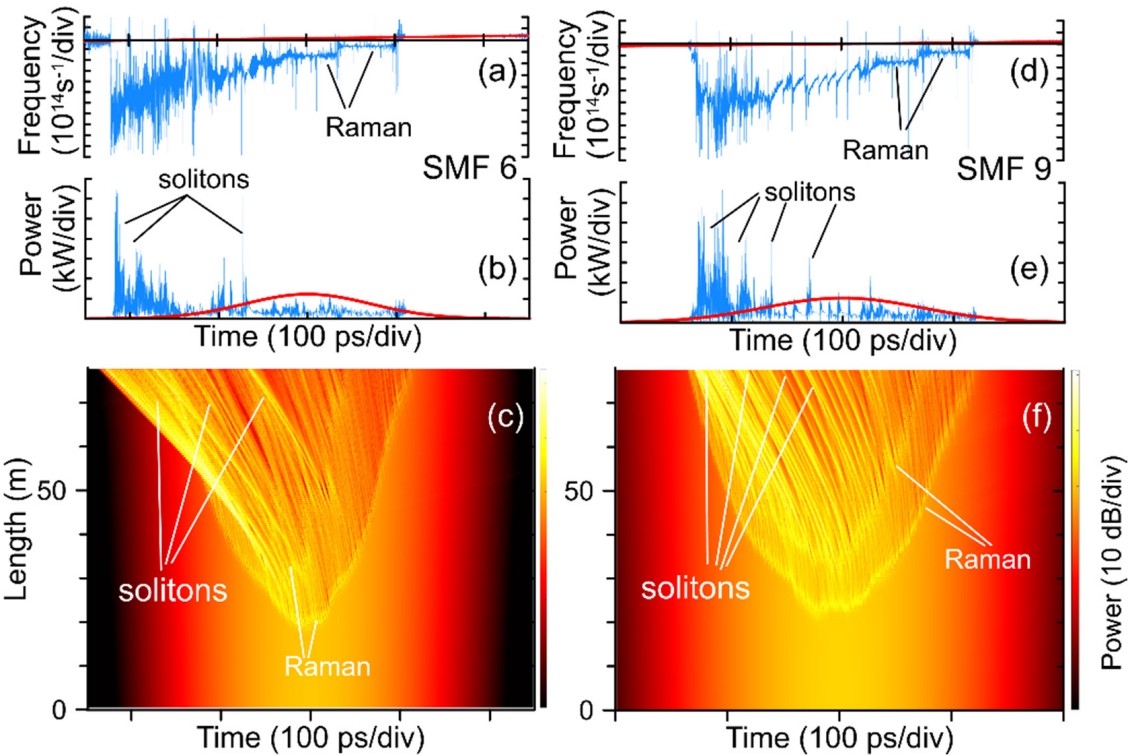

**Figure 6.** Frequency modulation of radiation at the input (red) and output (blue lines) from SMF6 (**a**) and SMF9 (**d**) fibers. Time envelopes the initial pulse (red) and output radiation (blue lines) from SMF6 (**b**) and SMF9 (**e**) fibers. The evolution of the pulse power during the propagation of radiation along the fiber SMF6 (**c**) and SMF9 (**f**).

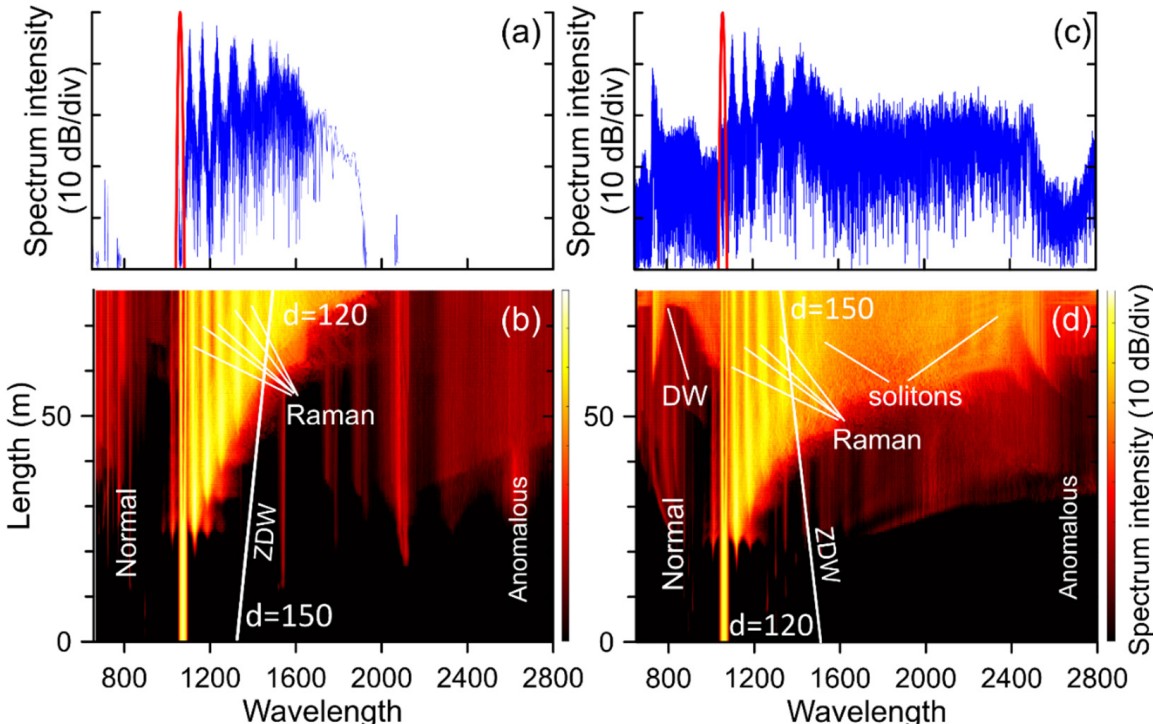

**Figure 7.** Spectra at the input (red) and output (blue lines) from fibers with longitudinally decreasing (**a**) and increasing (**c**) diameters. The evolution of the spectral density during the propagation of radiation through the fiber with longitudinally decreasing (**b**) and increasing (**d**) diameter.

In comparing the spectral evolution in the fibers with longitudinally decreasing and increasing diameters (Figure 7), we note that in the first case (when the diameter decreases along the length from 150 to 120 μm), ZDW shifts towards longer wavelengths, which leads to a gradual broadening of the normal dispersion region. As a result, only a small part of the spectrum energy is transferred to the region of anomalous dispersion, and this energy is insufficient for the formation of solitons with high peak power and a wide spectrum. On the contrary, in the case of the fiber with a longitudinally increasing diameter, ZDW shifts to the left, increasing the region of anomalous dispersion. Thus, intense radiation, initially located in the region of normal dispersion, can approach ZDW and move into the region of anomalous dispersion, forming a wide soliton spectrum. As a result, the simulation shows that the output spectrum obtained in a fiber with an increasing diameter is significantly wider than in a fiber with a decrease in diameter. These conclusions are in agreement with the experiment. In addition, the presence of an intense soliton component in the spectrum allows the appearance of dispersion waves (DW) (Figure 7d), the generation condition of which is phase matching between the soliton and dispersive radiation. As can be seen, the formation of the dispersive radiation band makes it possible to achieve a significant expansion of the short-wavelength boundary of the SC.

Simulations in the temporal domain show results similar to those previously obtained for the fibers with constant diameters (Figure 8). One can notice the traces of multiple Raman scattering and the formation of a large number of ultrashort solitons. The important difference between the fibers with longitudinally decreasing and increasing diameters is the maximal peak power of solitons at the fiber output. In the second case, when ZDW shifts towards shorter wavelengths, the soliton formation occurs earlier than in the first case, providing better soliton compression and higher peak power. Ultimately, it leads to a greater output spectrum broadening into the long wavelength range.

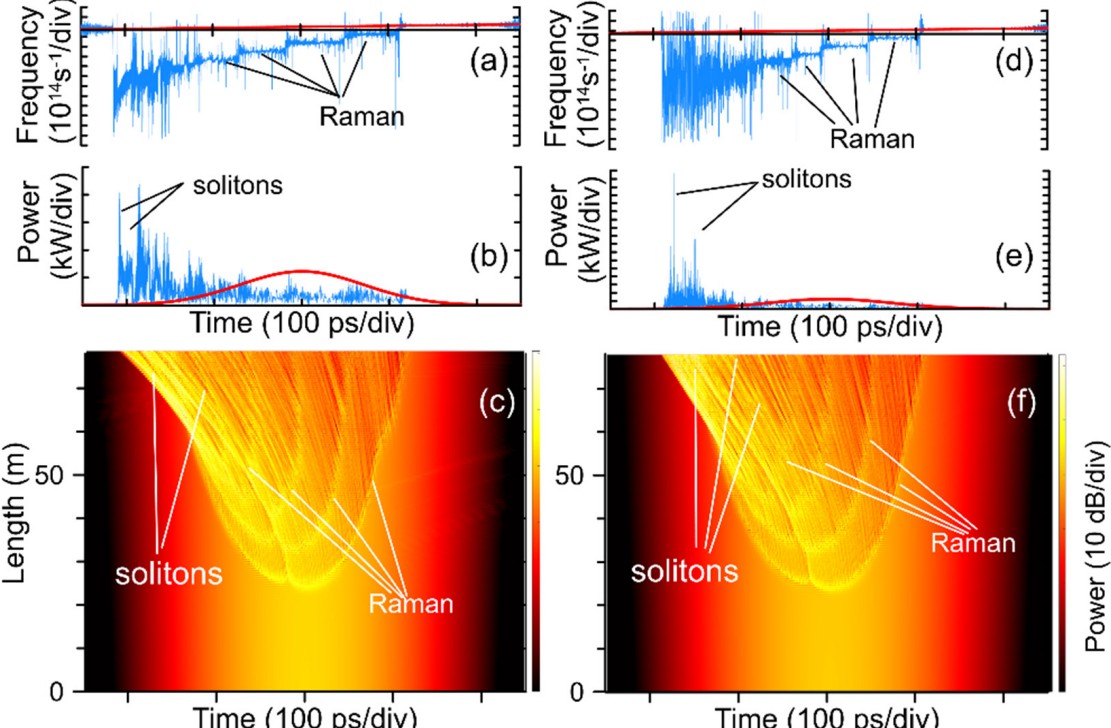

**Figure 8.** Frequency modulation of radiation at the input (red) and output (blue lines) from fibers with longitudinally decreasing (**a**) and increasing (**d**) diameters. Time envelopes of the initial pulse (red) and output radiation (blue lines) from fibers with longitudinally decreasing (**b**) and increasing (**e**) diameters. The evolution of the pulse power during the propagation of radiation through the fibers with longitudinally decreasing (**c**) and increasing (**f**) diameters.

## 4. Conclusions

The spectral conversion of sub-nanosecond pulses of the spectral range of 1 μm in optical fibers with a DDF has been studied experimentally and theoretically. After the introduction of radiation into a nonlinear medium, SC generation was obtained with the maximum achieved broadening of the optical spectrum up to 1496 nm at a signal level of −30 dB. The average maximum output power was 340 mW. The durations of the pulse envelopes were also obtained, which amounted to approximately 1 ns.

It is shown that, in comparison with standard optical fibers, the spectral broadening in the samples of the studied fibers is greater by 200 nm. The difference between the spectral transformation of radiation depending on the direction of propagation is also demonstrated. To confirm the obtained experimental data, a numerical simulation of radiation propagation over several fiber samples was carried out: DDF, SMF6 and SMF9. As can be seen from the presented simulation, the main nonlinear processes in the generation of SC in the DDF sample are the multiple SRS and the generation of ultra-short Raman solitons. This combination of processes makes it possible to obtain an octave-band SC, and varying the direction of radiation propagation in DDF makes it possible to control in which spectral region the most efficient conversion will occur.

**Author Contributions:** Conceptualization, V.A.K., V.B.T. and A.A.F.; methodology, I.V.Z., D.A.K., A.A.S. and V.A.K.; investigation, I.V.Z., D.A.K., A.S.A. and V.A.K.; data analysis, I.V.Z., D.A.K., A.S.A., A.A.S., V.A.K. and V.B.T.; writing—original draft preparation and editing, I.V.Z., V.A.K., D.A.K., A.S.A., A.A.F., A.A.S. and V.B.T.; visualization, I.V.Z., V.A.K., D.A.K. and A.S.A.; numerical simulation, D.A.K., A.S.A. and A.A.F.; resources, A.A.S.; supervision, V.A.K.; and project administration, V.B.T. All authors have read and agreed to the published version of the manuscript.

**Funding:** The work of I.V.Z., V.A.K. and V.B.T. was funded with the financial support of the Ministry of Science and Higher Education of the Russian Federation (#№ 075-15-2022-315), carried out on the basis of the World-Class Research Center «Photonics». D.A.K. and A.S.A. are grateful for the support from the Ministry of Science and Higher Education of the Russian Federation (project #075-15-2021-581) and the Russian Science Foundation (#22-72-10072). A.A.S. thanks for the support of the Russian Science Foundation (#22-12-00396). A.A.F. is supported by the European Union's Horizon 2020 research and innovation program (H2020-MSCA-IF-2020, #101028712).

**Institutional Review Board Statement:** Not applicable.

**Informed Consent Statement:** Informed consent was obtained from all subjects involved in the study.

**Data Availability Statement:** The data supporting the findings of this study are available within the article.

**Conflicts of Interest:** The authors declare no conflict of interest.

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
