# Peer review of "Broadband Supercontinuum Generation in Dispersion Decreasing Fibers in the Spectral Range 900–2400 nm"

_photonics, doi:10.3390/photonics9100773_

Round 1

Reviewer 1 Report

The authors demonstrate and study spectrally flat supercontinuum generation in the wavelength range of 900-2400 nm in silica-based fibers of variable core diameter and dispersion. Technical content of the manuscript is rather solid and relevant to the Journal's scope. However, the paper has a few minor flaws which should be fixed prior the final acceptance. I suggest to properly address the following concerns:

11.       In the Introduction, the listed application areas which drive practical research interest into supercontinuum generation is incomplete. Historically, the most important and motivating application field was the time-frequency metrology using ultra-stable femtosecond optical frequency combs. They also were spectrally broadened to more than an octave in silica-based highly-nonlinear fibers including those with varied core diameter and dispersion, for example, as shown in the work: https://doi.org/10.1070/QE2014v044n06ABEH015458

22.       It is not explained how the temporal pulse envelopes shown in Fig. 2, 3, 4 were measured. If they were acquired by means of an ultra-fast photodiode and oscilloscope pair, then I wonder why the authors apply symmetric time scale for the evidently asymmetric pulse shapes when plotting their graphs?  

33.       According to the experimental optical spectra in Fig. 3., the fiber SMF6 is definitely worser than the fiber SMF9 in terms of spectral broadening towards longer wavelengths. According to the numerical simulation results in Fig.5, the difference between those fibers in reaching the 2000-nm spectral range is not so evident. Please, explain additionally those peculiarities of experimental and theoretical results.

44.       When comparing the SC spectra in Fig. 4 (b) and Fig. 4(c), it is hardly possible to find significant difference in the spectral broadening despite the significant (double) difference in the average powers of seeding radiation. How such “saturation” of spectral broadening can be explained?

Thanks

Author Response

  1.       In the Introduction, the listed application areas which drive practical research interest into supercontinuum generation is incomplete. Historically, the most important and motivating application field was the time-frequency metrology using ultra-stable femtosecond optical frequency combs. They also were spectrally broadened to more than an octave in silica-based highly-nonlinear fibers including those with varied core diameter and dispersion, for example, as shown in the work: https://doi.org/10.1070/QE2014v044n06ABEH015458

Answer: Thank you for your comment, the introduction has been expanded.

  1.     It is not explained how the temporal pulse envelopes shown in Fig. 2, 3, 4 were measured. If they were acquired by means of an ultra-fast photodiode and oscilloscope pair, then I wonder why the authors apply symmetric time scale for the evidently asymmetric pulse shapes when plotting their graphs?  

Answer: An estimate of the pulse envelope duration was obtained from the FWHM. The time scales were also changed.

  1.       According to the experimental optical spectra in Fig. 3., the fiber SMF6 is definitely worser than the fiber SMF9 in terms of spectral broadening towards longer wavelengths. According to the numerical simulation results in Fig.5, the difference between those fibers in reaching the 2000-nm spectral range is not so evident. Please, explain additionally those peculiarities of experimental and theoretical results.

Answer: As mentioned in the article, the numerical simulation was simplified and did not take into account the optical bending losses of the fiber. Since initially the DDF fiber was on a coil of a certain diameter, the other samples were about the same diameter. This additionally contributes to the difference between the experimental and theoretical data obtained.

  1.       When comparing the SC spectra in Fig. 4 (b) and Fig. 4(c), it is hardly possible to find significant difference in the spectral broadening despite the significant (double) difference in the average powers of seeding radiation. How such “saturation” of spectral broadening can be explained?

Answer: In the spectral region above 1.8 μm, the power fraction increased with increasing pumping power (Fig.4(b) shows 47% in the spectral region of 1.8-2.4 μm (red spectrum), Fig.4(c) shows 52% (red spectrum). The "saturation" in the long-wave boundary is not due to optical losses in the used material itself.

Reviewer 2 Report

This paper proposed a broadband supercontinuum generation in dispersion decreasing fibers in the spectral range 900-2400 nm. This paper is well organized and its presentation is good. However, some minor issues still need to be improved:

1. How much pump energy is injected into the DDF fiber or SMF fiber.
2. In page 5, line 153, how to derive from the coherent continuum about pump symmetry, and in which parts of the supercontinuous spectrum is reflected in the anti-noise (spontaneous emission noise) performance of the supercontinuous spectrum

3. In previous understanding, supercontinuum spectra over an octave required the pumped light in negative dispersion regions, but this article seems to be different from this understanding. How does the pumped light achieve wider than one octave in positive dispersion? What are the physical reasons behind it?

4. In page 9, when the dispersive wave moves farther away from the pumped light, the spectrum is narrower. The article indicates the reason is the lack of energy. Whether increasing energy can get the result which is more consistent with previous experience? The transmission distance has a great influence on the dispersion wave, what is the reason?

5. In figure7(d), can the dispersive waves around 800nm that appear in the simulation be detected in the experiment?

6. The author should check English in entire paper.

I hope the authors can revise the manuscript carefully according to the above issues.

Author Response

1. How much pump energy is injected into the DDF fiber or SMF fiber.

Answer: The input peak power was 1-2 kW, and the energy in the input pulse was 0.8 µJ (800 nJ). 

2. In page 5, line 153, how to derive from the coherent continuum about pump symmetry, and in which parts of the supercontinuous spectrum is reflected in the anti-noise (spontaneous emission noise) performance of the supercontinuous spectrum

Answer: This was part of a literary overview and has now been moved to the beginning of the manuscript so as not to confuse the reader (line 61-62).

3. In previous understanding, supercontinuum spectra over an octave required the pumped light in negative dispersion regions, but this article seems to be different from this understanding. How does the pumped light achieve wider than one octave in positive dispersion? What are the physical reasons behind it?

Answer: As can be seen, two nonlinear processes, Raman scattering in the normal dispersion region and the formation of ultrashort pulses (solitons) in the anomalous dispersion region, are mainly involved in the changes in the pulse envelope. The process of envelope breaking (optical wave breaking) begins near the top of the pulse, where nonlinear effects are the strongest. Initially, there is a Raman shift to the region of longer waves. Two consecutive Raman cascades can be seen in the figure. In the output pulse, traces of this process are noticeable near the rear (right) edge of the pulse as areas of flat frequency modulation. Further, when part of the pulse envelope moves into the region of anomalous dispersion (compare with Fig. 5), solitons are formed, shifting into the range of even longer waves. In the output pulse, the group of solitons with the maximum peak power is shifted to the leading (left) front. The frequency modulation (Fig. 6 (a, d)) shows that spectrally this group is shifted to the region of maximum wavelengths, which agrees with the results of the spectral evolution simulation (Fig. 5). 

4. In page 9, when the dispersive wave moves farther away from the pumped light, the spectrum is narrower. The article indicates the reason is the lack of energy. Whether increasing energy can get the result which is more consistent with previous experience? The transmission distance has a great influence on the dispersion wave, what is the reason?

Answer:  We note that the condition for the generation of dispersive radiation is the occurrence of phase synchronism between the soliton and the dispersive wave [41]. Comparing Figs. 7(b) and (d) in the field of evolution of the dispersive wave spectrum, we can see that, in the fiber with increasing diameter (d), the soliton generation occurs earlier. Accordingly, the soliton component of the spectrum arising in this case is much wider than in the fiber with decreasing diameter (b). As a result, this leads to an increase in the possible points of phase synchronism, i.e., points of energy transfer from the soliton part of the spectrum to the dispersive emission. Undoubtedly, increasing the energy of the initial pulse will also lead to a broadening of the soliton part of the spectrum and, consequently, to greater energy pumping from it into the dispersive emission spectrum.  

5. In figure7(d), can the dispersive waves around 800nm that appear in the simulation be detected in the experiment?

Answer: As we noted, the exact description of the experiment during modeling was not part of our task. The main task was to describe the physical mechanisms acting during pulse propagation and to find out the features associated with the generation of the supercontinuum in a longitudinally nonuniform fiber. We also noted that a necessary condition for the generation of dispersive radiation is the occurrence of phase synchronism between the soliton and the dispersive wave [41]. In order to generate a wider band of dispersive radiation with the used fibers, an increase in the energy of the propagating pulse can be used (in particular, this is evidenced in Fig. 4). In a broader sense, one option could be the use of fibers with a specially calculated longitudinal dispersion profile, which provides for the appearance of phase-synchronism points between the generated solitons and dispersive radiation with a special wavelength (e.g., 800 nm). 

6. The author should check English in entire paper.

Answer: Thanks for the comment, the manuscript was checked agains

Reviewer 3 Report

see the attachment

Author Response

1. What is the type of the optical spectrum analyzer and the oscilloscope? Why the y-axis coordinates of Fig. 2(a) and (b) are not dBm? 

Answer:  In the manuscript, the spectrum in logarithmic scale with relative intensity is presented, for uniformity of all presented spectra. As a spectrum analyzer was used HP (spectral resolution 0.1 nm with a range of 700-1700 nm) and Yokogawa (spectral resolution 0.1 nm with a range of 1200-2400 nm). The various photodetectors for different spectral regions 0.8-1.7 µm (rise time < 35 ps) and 1.5–2.5 µm (rise time < 100 ps), and a Tektronix oscilloscope (bandwidth up to 4 GHz) were used for pulses envelope measurement. 

2. The author mentioned “Precisely, a significant increase in the peak intensity at wavelengths of 1040 and 1120 nm due to SRS with a transformation in the region of 1063 nm (Figure 2 (b)) was caused by the SPM effect.” why it is caused by SPM effect? 

Answer: In the resonator we have a normal GVD mode together with subpicosecond pulses, which leads to broadening dynamics due to the interaction of the FSM and the normal fiber GVD [1]*.  The sentence in question meant "widening", not "intensity". This is a typo.

 [1]*Agrawal P. Nonlinear Fiber Optics Fourth Edition. 

3. The author mentioned “In both cases, the width was estimated by a signal level 133 of -30 dB.” the - 30 dB should be labeled in Fig. 3. 

Answer: Thank you for the comment. Changes have been made to the manuscript. A -30 dB in our case equals 10^-3 level and this is noted in the text and lines added to the figures.

4. “1111 nm (8998 cm1), 1163 nm (8598 cm1), 1220 nm (8198 cm1)),” should be “1111 nm (8998 cm-1 ), 1163 nm (8598 cm-1 ), 1220 nm (8198 cm-1 )),” 

Answer: thanks for the comment. Changes have been made to the manuscript.

5. A comparison with reported works should be presented to demonstrate the advantages and disadvantages of the proposed work

Answer: If we compare our results with other all-fiber silica-based supercontinuum generators, we are the first to demonstrate the possibility of generation in different spectral regions due to the orientation of the DDF fiber. Also, unlike many other systems, we have quite small peak power input to NF for SC generation (1-2 kW against tens of kW) [1,2* ], and the system is fairly simple and without complex elements ( for example, in this paper, to get a fully fiber circuit, the authors used a mode field adapter (MFA) [3*]).  If we compare with works using GeO2 fibers [19-22 in manuscript], however, in our case we go to the short-wave region of the spectrum due to the use of MO emitting in the 1 μm region and the taper of the DDF. To increase the energy in the pulse, further optimization of the studied system is required, as well as a more detailed analysis of the temporal dynamics of the pulses.Changes have been made to the manuscript.

1* Krupa, K.; Louot, C.; Couderc, V.; Fabert, M.; Guenard, R.; Shalaby, B.M.; Tonello, A.; Pagnoux, D.; Leproux, P.; Bendahmane, A.; et al. Spatiotemporal Characterization of Supercontinuum Extending from the Visible to the Mid-Infrared in a Multimode Graded-Index Optical Fiber. Opt. Lett. 2016, 41, 5785.doi:10.1364/ol.41.005785 

2* TeÄŸin, U.; Ortaç, B. Cascaded Raman Scattering Based High Power Octave-Spanning Supercontinuum Generation in Grad-ed-Index Multimode Fibers. Sci Rep 2018, 8. doi:10.1038/s41598-018-30252-9

3* Qi, X.; Chen, S.; Li, Z.; Liu, T.; Ou, Y.; Wang, N.; Hou, J. High-Power Visible-Enhanced All-Fiber Supercontinuum Generation in a Seven-Core Photonic Crystal Fiber Pumped at 1016 Nm. Opt. Lett. 2018, 43, 1019. doi:10.1364/ol.43.001019

Reviewer 4 Report

The authors experimentally and numerically studied more than octave spanning supercontinuum (SC) generation in dispersion decreasing fibers (DDFs). The paper fits very well to the scopes of Photonics. The experimental results are interesting and seem reliable.
The main concern is the numerical part of the paper.

1. Why do the authors not take into account frequency-dependent linear losses for modeling SC generation? It is well known that at wavelengths >2 μm, the losses increase sharply, and at fiber lengths of several tens of meters, when the spectrum expands significantly beyond 2 μm, it does not seem correct to neglect these losses.

2. The authors do not take into account the frequency dependence of gamma, which can lead to significant errors for SC widths greater than an octave.

3. In Figure 6, the authors marked supposedly solitons. It is not clear how exactly the authors determined that these parts were solitons. The figure shows complex time structures and it is not obvious that the indicated components are solitons. Probably, the authors can try to build spectrograms, from which this will be clear.

4. It is also not clear how the authors determined solitons in Fig. 7. Maybe the marked components arose as a result, for example, of interactions between other spectral components and are not solitons at all?

5. Paragraph 3 in the introduction (starting from line 43) is not very carefully  written and needs to be improved, to my opinion.
5.1. The authors state “…fibers with a high nonlinearity coefficient are increasingly used to generate SCs such as ZBLAN… [14]”. ZBLAN fibers in the general case and, in particular, in [14] are not highly nonlinear ones. For them, n2 is slightly smaller than n2 for silica glass. In [14], the mode areas are comparable with the SMF28e mode area; accordingly, the Kerr coefficients are comparable with the coefficient for SMF28e.
5.2. The authors mention fluoride and ZBLAN fibers separately, although ZBLAN belongs to the fluoride class.
5.3 It is not clear why when mentioning SC in chalcogenide fibers, the authors cite only [17]. It would be more logical to cite either a review on SC in chalcogenide fibers or an original work where SC with some outstanding characteristics was obtained for the first time. Article [17] is just one of the papers on the topic.
5.4. The authors list “Nowadays, fibers with a high nonlinearity coefficient are increasingly used to generate SCs, such as ZBLAN (ZrF4–BaF2–LaF3–AlF3–NaF) [14], photonic crystal fibers (PCFs) [15], fluoride [16] or chalcogenide [17]”. PCFs define the structure of the fiber and can be made of different glasses, while the rest fibers in this sentence refer to the type of glass. In addition, there are also tellurite fibers, which are widely used for SC generation.
5.5. When the authors talk about the generation of SCs in GeO2-doped fibers, they cite [18], where the red boundary is only 2.2 μm, although there are experimental works by different groups in which SCs with red boundaries of more than 3 μm were obtained in GeO2-doped fibers (and not only [19]).
5.6. Further, “…one of the types of which are decreasing dispersion fibers (DDF) [20–22]”. As far as I understand, DDF was not used in [21].
So, I recommend the authors to revise the paragraph 3 in the introduction taking into account the comments and to cite papers more carefully.

Author Response

  1. Why do the authors not take into account frequency-dependent linear losses for modeling SC generation? It is well known that at wavelengths >2 μm, the losses increase sharply, and at fiber lengths of several tens of meters, when the spectrum expands significantly beyond 2 μm, it does not seem correct to neglect these losses.
  2. The authors do not take into account the frequency dependence of gamma, which can lead to significant errors for SC widths greater than an octave.

Answer: We agree with the distinguished reviewer that the frequency dependences of the loss coefficients and nonlinearity are extremely important for an accurate description of the experiment. At the same time, as we noted, the exact description of the experiment was not part of our task. The main task was to describe the physical mechanisms acting during pulse propagation and to elucidate the features associated with the generation of the supercontinuum in the longitudinally nonuniform fiber.

We also pointed out that the modeling of supercontinuum generation within the framework of the task requires significant computational costs, as it involves consideration of large ranges in both frequency and time domains (218 grid points). In these circumstances, we have neglected a number of factors, such as the shape and exact frequency modulation of the initial pulse, the frequency dependences of the loss factors and nonlinearity, whose consideration does not lead to fundamental changes in the character of the pulse evolution, but can complicate and delay the calculations. Changes have been made in the manuscript.

  1. In Figure 6, the authors marked supposedly solitons. It is not clear how exactly the authors determined that these parts were solitons. The figure shows complex time structures and it is not obvious that the indicated components are solitons. Probably, the authors can try to build spectrograms, from which this will be clear.

Answer: To answer the question, we have additionally placed figures illustrating the evolution of the pulse in the time domain (Fig. 6 (c, f)). As can be seen, two nonlinear processes, Raman scattering in the normal dispersion region and the formation of ultrashort pulses (solitons) in the anomalous dispersion region, are mainly involved in the changes in the pulse envelope. The process of envelope breaking (optical wave breaking) begins near the top of the pulse, where nonlinear effects are the strongest. Initially, there is a Raman shift to the region of longer waves. Two consecutive Raman cascades can be seen in the figure. In the output pulse, traces of this process are noticeable near the rear (right) edge of the pulse as areas of flat frequency modulation. Further, when part of the pulse envelope moves into the region of anomalous dispersion (compare with Fig. 5), solitons are formed, shifting into the range of even longer waves. In the output pulse, the group of solitons with the maximum peak power is shifted to the leading (left) front. The frequency modulation (Fig. 6 (a, d)) shows that spectrally this group is shifted to the region of maximum wavelengths, which agrees with the results of the spectral evolution simulation (Fig. 5).  Changes have been made to the text.

  1. It is also not clear how the authors determined solitons in Fig. 7. Maybe the marked components arose as a result, for example, of interactions between other spectral components and are not solitons at all?

Answer: Similarly to the previous one, the results of simulation of pulse propagation in the DDF in the time domain were additionally placed (Fig. 8). Changes have been made to the text.

  1. Paragraph 3 in the introduction (starting from line 43) is not very carefully  written and needs to be improved, to my opinion.

5.1. The authors state “…fibers with a high nonlinearity coefficient are increasingly used to generate SCs such as ZBLAN… [14]”. ZBLAN fibers in the general case and, in particular, in [14] are not highly nonlinear ones. For them, n2 is slightly smaller than n2 for silica glass. In [14], the mode areas are comparable with the SMF28e mode area; accordingly, the Kerr coefficients are comparable with the coefficient for SMF28e.

5.2. The authors mention fluoride and ZBLAN fibers separately, although ZBLAN belongs to the fluoride class.

Answer: Thank you for the comment. Changes have been made to the manuscript. A complete review on Infrared fibers [1*] has been added (in manuscript [15]).

[1]*Tao G. et al. Infrared fibers // Adv. Opt. Photonics. 2015. Vol. 7, â„– 2. https://doi.org/10.1364/AOP.7.000379

5.3 It is not clear why when mentioning SC in chalcogenide fibers, the authors cite only [17]. It would be more logical to cite either a review on SC in chalcogenide fibers or an original work where SC with some outstanding characteristics was obtained for the first time. Article [17] is just one of the papers on the topic.

Answer: Thanks for the remark. The [17] has been replaced by [2]* (in manuscript [18])

[2]*Dai, S.; Wang, Y.; Peng, X.; Zhang, P.; Wang, X.; Xu, Y. A Review of Mid-Infrared Supercontinuum Generation in Chalcogenide Glass Fibers. Appl. Sci. 2018, 8, 707.  https://doi.org/10.3390/app8050707

5.4. The authors list “Nowadays, fibers with a high nonlinearity coefficient are increasingly used to generate SCs, such as ZBLAN (ZrF4–BaF2–LaF3–AlF3–NaF) [14], photonic crystal fibers (PCFs) [15], fluoride [16] or chalcogenide [17]”. PCFs define the structure of the fiber and can be made of different glasses, while the rest fibers in this sentence refer to the type of glass. In addition, there are also tellurite fibers, which are widely used for SC generation.

Answer: Thank you for the remark. Tellurite fibers have been added to the manuscript [3]*.

[3]*T. Sylvestre, E. Genier, A. N. Ghosh, P. Bowen, G. Genty, J. Troles, A. Mussot, A. C. Peacock, M. Klimczak, A. M. Heidt, J. C. Travers, O. Bang, and J. M. Dudley, "Recent advances in supercontinuum generation in specialty optical fibers [Invited]," J. Opt. Soc. Am. B 38, F90-F103 (2021) https://doi.org/10.1364/JOSAB.439330

5.5. When the authors talk about the generation of SCs in GeO2-doped fibers, they cite [18], where the red boundary is only 2.2 μm, although there are experimental works by different groups in which SCs with red boundaries of more than 3 μm were obtained in GeO2-doped fibers (and not only [19]).

Answer: The article [18] is cited because it compares GeO2 with SMF-28. The article [4, 5]* (in manuscript [22,23]) was added.

[4]*Anashkina, E.A.; Andrianov, A.V.; Koptev, M.Y.; Muravyev, S.V.; Kim, A.V. Towards Mid-Infrared Supercontinuum Generation With Germano-Silicate Fibers. IEEE J. Select. Topics Quantum Electron. 2014, 20, 643–650. 10.1109/JSTQE.2014.2321286

[5]*X. Wang et al., "High-Power All-Fiber Supercontinuum Laser Based on Germania-Doped Fiber," in IEEE Photonics Technology Letters, vol. 33, no. 23, pp. 1301-1304, 1 Dec.1, 2021, doi: 10.1109/LPT.2021.3117283.

5.6. Further, “…one of the types of which are decreasing dispersion fibers (DDF) [20–22]”. As far as I understand, DDF was not used in [21].

Answer: Thank you for the comment. Changes have been made to the manuscript.

Round 2

Reviewer 4 Report

The authors did a great job of improving the manuscript, gave very detailed answers and comments to all my questions and suggestion, made appropriate corrections to the text, and added new numerically modeled figures proving statements in the paper. Therefore, I am happy to recommend the paper for publication in Photonics.